# Task-Rate-Related Neural Dynamics Using Wireless EEG to Assist Diagnosis and Intervention Planning for Preschoolers with ADHD Exhibiting Heterogeneous Cognitive Proficiency

**DOI:** 10.3390/jpm12050731

**Published:** 2022-04-30

**Authors:** I-Chun Chen, Chia-Ling Chen, Chih-Hao Chang, Zuo-Cian Fan, Yang Chang, Cheng-Hsiu Lin, Li-Wei Ko

**Affiliations:** 1International Ph.D. Program in Interdisciplinary Neuroscience, College of Biological Science and Technology, National Yang Ming Chiao Tung University, Hsinchu 30010, Taiwan; cyc@tmu.edu.tw; 2Department of Physical Medicine and Rehabilitation, Ton-Yen General Hospital, Hsinchu 30268, Taiwan; 3Department of Physical Medicine and Rehabilitation, Chang Gung Memorial Hospital, Linkou, Taoyuan 33305, Taiwan; 4Graduate Institute of Early Intervention, College of Medicine, Chang Gung University, Taoyuan 33302, Taiwan; 5Department of Biological Science and Technology, National Yang Ming Chiao Tung University, Hsinchu 30010, Taiwan; changchihao819@gmail.com (C.-H.C.); jkevin851117@gmail.com (Z.-C.F.); 6Brain Research Center and the Center for Intelligent Drug Systems and Smart Bio-Devices (IDS2B), National Yang Ming Chiao Tung University, Hsinchu 30010, Taiwan; yanglalala@gmail.com; 7Institute of Bioinformatics and Systems Biology, National Yang Ming Chiao Tung University, Hsinchu 30010, Taiwan; 8Doctor Lin’s Psychiatric Clinic, Hsinchu 30069, Taiwan; hu_1220@yahoo.com.tw; 9Department of Electronics and Electrical Engineering, Institute of Electrical and Control Engineering, National Yang Ming Chiao Tung University, Hsinchu 30010, Taiwan; 10Drug Development and Value Creation Research Center, Department of Biomedical Science and Environmental Biology, Kaohsiung Medical University, Kaohsiung 80708, Taiwan

**Keywords:** ADHD, Conners Kiddie Continuous Performance Test (K-CPT), preschoolers, wireless electroencephalography, cognitive proficiency

## Abstract

This study used a wireless EEG system to investigate neural dynamics in preschoolers with ADHD who exhibited varying cognitive proficiency pertaining to working memory and processing speed abilities. Preschoolers with ADHD exhibiting high cognitive proficiency (ADHD-H, *n* = 24), those with ADHD exhibiting low cognitive proficiency (ADHD-L, *n* = 18), and preschoolers with typical development (TD, *n* = 31) underwent the Conners’ Kiddie Continuous Performance Test and wireless EEG recording under different conditions (rest, slow-rate, and fast-rate task). In the slow-rate task condition, compared with the TD group, the ADHD-H group manifested higher delta and lower beta power in the central region, while the ADHD-L group manifested higher parietal delta power. In the fast-rate task condition, in the parietal region, ADHD-L manifested higher delta power than those in the other two groups (ADHD-H and TD); additionally, ADHD-L manifested higher theta as well as lower alpha and beta power than those with ADHD-H. Unlike those in the TD group, the delta power of both ADHD groups was enhanced in shifting from rest to task conditions. These findings suggest that task-rate-related neural dynamics contain specific neural biomarkers to assist clinical planning for ADHD in preschoolers with heterogeneous cognitive proficiency. The novel wireless EEG system used was convenient and highly suitable for clinical application.

## 1. Introduction

Attention deficit hyperactivity disorder (ADHD) is the most prevalent childhood-onset neuropsychiatric disorder, the symptoms typically emerging early (at preschool age) and persisting into adulthood [1,2,3]. In 2011, the American Academy of Pediatrics updated the ADHD clinical practice guidelines, recommending clinicians evaluate for ADHD in all children from the ages of 4–5 years [4]. Identification of preschool children at risk for ADHD and providing early intervention are imperative [3,5,6,7]. The diagnosis of preschool ADHD is challenging due to the relative scarcity of clinical interviews and rating scales for preschoolers, as well as the variable nature of structure across preschool settings, which makes them unfavorable for multi-informant assessments [8,9]. To facilitate diagnosis, the continuous performance test (CPT) and electroencephalography (EEG), which have the advantages of safety, low price, quick preparation, wide availability, and convenience, can be used to directly investigate individual performance on attention-demanding tasks [10,11,12,13,14,15].

Regarding the evaluation of a brain-based biomarker in neurodevelopmental disorders, several criteria should be considered in terms of method, such as practicability, time-efficiency of data acquisition, cost-effectiveness, and neurobiological interpretability, which indicates the neural markers should be grounded in pathophysiological mechanisms of atypical development [10]. EEG is increasingly viewed as the modality of interest through which biomarkers for the characterization of neurodevelopmental disorders can be identified [10,11,12,13,14,16,17]. Power spectral density (PSD) studies on ADHD have revealed one consistent finding: individuals with ADHD manifest high absolute and relative theta power and theta/beta ratio (TBR) in the frontocentral region of the brain at rest [11,12,14,16,18,19,20]. However, Clarks, et al. [21] proposed three distinct EEG clusters to define children with ADHD, involving increased slow-wave activity and deficiencies in the fast wave, increased high-amplitude theta with deficiencies in beta activity, and an excess of beta. Furthermore, one meta-analysis, including nine studies with a total of 1253 children/adolescents with ADHD and 517 without ADHD, concluded excessive TBR cannot be considered a reliable diagnostic measure [22]. As for the other frequency power bands, studies have reported that delta power is typically higher in individuals with ADHD than in healthy controls (HCs) [12,17,21,23,24,25], whereas alpha and beta power are usually lower [12,16,17,21,23].

The cognitive proficiency index (CPI) in the Wechsler Preschool and Primary Scale of Intelligence–Fourth Edition (WPPSI-IV) measures the child’s abilities in working memory and processing speed, and provides an estimate of the efficiency with which cognitive information is processed in learning, problem-solving, and higher-order reasoning [26]. Several studies have identified that school-aged children with ADHD show poorer performance in working memory and processing speed than in perceptual and verbal functional domains when compared with their peers of average intelligence [27,28,29,30,31,32]. Regarding preschool children, the deficits in varied executive functioning, including inhibition control, working memory, speed of processing, and planning/organization, may be general markers of ADHD [33,34,35,36,37,38]. However, ADHD appears to be characterized by neurobehavioral heterogeneity both in preschool and school-age children; the subgroup of youths with ADHD may perform at a high level (as high as their neurotypical counterparts) on related tasks [32,34,39].

With regard to ADHD heterogeneity, most studies have used the subtypes outlined in the *Diagnostic and Statistical Manual of Mental Disorders, 4th Edition*, or EEG profile clusters for grouping, computing, and comparing the resting EEG band power of subpopulations of children with ADHD [21,23,40,41,42]. To the best of our knowledge, few studies have explored the neural dynamics of subgroups of preschoolers with ADHD according to cognitive proficiency by manipulating task rate.

Most EEG studies on children with ADHD have used wet-electrode systems with a stretch cap. In the present study, a wireless wearable EEG headset was used to quantify brain dynamics during CPT. Subsequently, we investigated EEG profiles under rest, slow-rate, and fast-rate task conditions among preschoolers with ADHD exhibiting high cognitive proficiency (ADHD-H), those with ADHD exhibiting low cognitive proficiency (ADHD-L), and those with typical development (TD). We hypothesized that EEG profiles during different conditions not only differentiated ADHD from TD but also reflected ADHD heterogeneity, specifically the varying levels of cognitive proficiency. Furthermore, the EEG profiles in the present study may contain specific neural biomarkers that can assist clinical planning for preschoolers with ADHD. For instance, in some cases without complete neurocognitive evaluating data, the neural dynamics measured by the easy-to-use system can provide clinicians with more information in diagnosis and treatment strategy.

## 2. Materials and Methods

Figure 1 illustrates the overall framework of the experiment, including subject evaluation, clinical diagnosis, EEG recording, signal processing, and data statistical analysis.

### 2.1. Participants

The participants comprised 73 preschoolers (aged 5–7 years), 42 of whom were diagnosed as having ADHD (34 boys and eight girls) and 31 of whom had TD (23 boys and eight girls). All the ADHD diagnoses were determined and confirmed in the clinical setting by the senior child and adolescent psychiatrists, based on the criteria of the DSM 5th Edition [1] and multiple perspectives, such as clinical interviews, neuropsychological tests, behavioral observations by qualified experienced psychologists in our institute, and behavioral rating scales obtained from parents and teachers. Those who had a history of brain disorders or any other neurological disorders, chromosomal or genetic disorders, autism spectrum disorder, learning disorder, or any other mental disorder were excluded. The study was implemented according to policies on human research and was approved by the Research Ethics Committee of the Taiwan National Health Research Institutes (EC1070401-F). The participants’ parents or guardians provided written informed consent to the academic use of the test results.

### 2.2. Apparatus

#### 2.2.1. Neuropsychological Measurements

##### Intelligence Quotient Test and Behavioral Rating Scales

Cognitive function was evaluated using the Taiwanese version of the Wechsler Preschool and Primary Scale of Intelligence–Fourth Edition (WPPSI-IV) [43], which was administered by qualified psychologists. We classified the participants with ADHD into groups with high and low cognitive proficiency according to their cognitive proficiency index (CPI) scores on the WPPSI-IV. Children with CPI scores of >85 and ≤85 were classified into the ADHD-H (*n* = 24) and ADHD-L (*n* = 18) groups, respectively. Two versions of the Disruptive Behavior Disorders Rating Scale (DBDRS) for parents and teachers [44] were used to evaluate ADHD symptoms.

##### Conners Kiddie Continuous Performance Test 2nd Edition

In this study, we used the 7.5-min Conners Kiddie Continuous Performance Test 2nd Edition (K-CPT2) [45] to assess attention-related performance. As displayed in Figure 2, the K-CPT2 involves 200 randomly presented trials divided into five blocks, each of which contains two 20-trial sub-blocks involving fast-rate and slow-rate tasks (1.5- and 3-s inter-stimulus interval [ISI], respectively). Superior to other commercial paradigms, the K-CPT features varied time intervals between stimuli, presenting a greater challenge for the subjects. At shorter ISIs, the stimuli occur and disappear quickly, and the subjects must process information rapidly. At longer ISIs, the stimuli are more persistent and last longer, but the subjects are required to maintain alertness and not let their attention wander. Throughout the entire K-CPT procedure, subjects must adjust and respond well depending on the varying task rate; that is, respond appropriately to the tasks with shorter and longer ISIs. Usually, in the context of children with ADHD, the reaction times may be longer (e.g., inattentive type) or shorter (e.g., hyperactive type) compared to healthy controls (HCs) in the shorter ISIs condition, while subjects may make more omission and commission errors in the longer ISIs condition [46]. The responses were used to compute scores reflecting various attention aspects.

The nine main standardized scores generated for each participant help assessors interpret the nature of their attention problems (through different aspects). Standardized T scores were defined as having a mean of 50 with a standard deviation (SD) of 10. Higher scores generally imply worse performance, except in special cases, such as when the hit reaction time (HRT) is measured. In these indexes, detectability (d’) indicates the ability of a participating child to discriminate nontargets from targets. Omissions refer to missed targets, and commissions signify incorrect responses to targets. Perseverations denote responses that are made in less than 100 ms following the stimulus. HRT measures the mean response speed of all nonperseverative responses over the entire test. The consistency of response speed was demonstrated by HRT SD and variability. The HRT block change computes the slope of the change in reaction time across five blocks of administration. The HRT ISI change refers to the slope of the change in reaction time between two ISIs; these nine main scores are all defined clearly in the K-CPT2 manual [45].

#### 2.2.2. Wireless and Wearable EEG

The apparatus used was a novel eight-channel wearable and portable EEG system [47] that consisted of a wireless EEG device and laptop computer (Figure 3). The wireless EEG acquisition device used eight semi-dry electrodes called hygroscopic sponge electrode sensors to record signals. The hygroscopic sponge sensors exhibited several superior characteristics [46]. First, the skin-contacting part was made of a soft sponge, which provided comfort during EEG acquisition. Second, the sensors worked with a small amount of water rather than the complex skin preparation and gel application process used in conventional wet-electrode systems. Third, the sensors can be assembled easily and rapidly, and fit properly into the original device. Fourth, the signal quality is comparable to that of NeuroScan, according to a study that revealed high average correlations of 90.03% and 82.56% in two-second and ten-second temporal resolutions, respectively, and 97.18% in frequency response [47]. In contrast to conventional wet-electrode systems, the users were able to monitor and obtain an individual’s EEG data to develop a real-time detection algorithm in a more natural and comfortable state during daily life. The aforementioned advantages make the wireless wearable system suitable for clinical use, particularly for young children.

### 2.3. Procedure and EEG Recording

Figure 4 illustrates the EEG recording procedure. To establish baseline resting power, resting-state EEG signals were recorded over 1 min with the participants’ eyes open. Subsequently, an instructor informed the participants of the experimental rules and provided them with the opportunity to practice prior to the official initiation of the K-CPT2. Raw EEG recording data were obtained from eight electrode sites on the scalp (Fp1, Fp2, Fz, C3, C4, Pz, O1, and O2) according to the standard international 10/20 system at a 1000-Hz sampling rate. In the present study, the linked earlobes were used as reference sites, and the impedance of each electrode was controlled below 100 k Ohm throughout the EEG recording session. The scalp–electrode impedance of the sponge electrodes used in our previous study [47] and the present study was higher (up to approximately 120 k Ohm) than that in the conventional wet-electrode system (typically approximately 20 k Ohm). However, in a circuit design containing an amplifier with an input impedance of approximately 200 M Ohm, a scalp–electrode impedance of up to 200 k Ohm is allowed for accurate signal acquisition with an approximate error rate of 0.1% [48].

### 2.4. Data Preprocessing and Processing

We used MATLAB software (MathWorks, Natick, MA, USA) to preprocess and process the EEG data. We extracted the data on rest- and task-related power for analysis. For artifact removal, first, 0.5 Hz high-pass and 50 Hz low-pass basic finite impulse response filters were applied to improve EEG signal quality. Second, EEG experts and the clinician identified and eliminated the components corresponding to muscle artifacts related to the participants’ restless movements during the experiment by independent component analysis (ICA) [46,49], and the signals were processed using ICLabel in EEGLAB toolbox, which provides automated independent component classification [50]. After the ICA process, the components of brain signals related to artifacts including eyes and muscle activities were eliminated. After the defective components were removed, the rest of the components were back-projected, and no further detected residual muscle artifacts related to fidgets remained in the data. These signals were used in further data analysis. According to the sequences of ISIs within sub-blocks (Figure 2), we further divided the task data into two epochs: 5 min of slow-task data comprising all segments with a 3-s ISI, and 2.5 min of fast-task data comprising all segments with a 1.5-s ISI. Four frequency bands were defined for spectral analyses: delta, theta, alpha, and beta (1–4, 4–8, 8–13, and 13–30 Hz, respectively). We determined the PSD of the EEG data using the short-time Fourier transform spectrogram function in the MATLAB signal processing toolbox. We analyzed the relative power, which was computed by dividing the absolute power in a designated frequency band by the sum of all the measured frequency bands and then multiplying the result by 100. Relative power is independent of bone thickness, skull resistance, and impedance variability; in addition, relative power is a more discriminative variable than absolute power in ADHD evaluation studies [51,52] (Figure 1).

### 2.5. Statistical Analysis

The demographic and neuropsychological test data of the three groups were compared using a one-way analysis of variance (ANOVA). Comparison analysis of categorical variables (e.g., sex) was conducted using Fisher exact tests.

Regarding EEG data analysis, the PSD under separate independent conditions (rest, slow-rate task, and fast-rate task) of two groups (all ADHD versus TD) as well as three groups (ADHD-H versus ADHD-L versus TD) were also compared using an independent *t*-test and a one-way ANOVA. To control type I error of possible false-positive results from multiple comparisons and to prevent type II error or false-negative error, we reduced the number of comparisons and spectral power was averaged by region as follows: prefrontal (Fp1, Fp2), central (C3, C4), and occipital (O1, O2). The power of specific frequency bands was compared in regions of interest (central and Pz). When at least 1 group showed statistically significant differences, pairwise comparisons were conducted. The false discovery rate correction was used for these post hoc analyses, in which the p-values were multiplied by the number of comparisons.

Additionally, to evaluate the group differences in alteration patterns in PSD under different conditions, we performed a generalized estimating equation, with the condition (rest, slow-rate task, and fast-rate task) as a within-subject factor and the group (ADHD-H, ADHD-L, and TD) as a between-subject factor. We computed the average power of specific frequency bands in all recording regions of the brain (prefrontal, Fz, central, Pz, and occipital). All the analyses were performed using IBM SPSS Statistics for Windows, version 22 (IBM Corp., Armonk, NY, USA).

## 3. Results

### 3.1. Neuropsychological Measurements

Table 1 presents the comparison of neuropsychological measurements between groups. The mean full-scale intelligence quotient of the ADHD-L group was lower than that of both the ADHD-H and TD groups (*p* = 0.019, *p* = 0.006, respectively); however, the verbal comprehensive index did not differ among the groups, which confirmed that all the participants completely comprehended the test instructions. CPI differed significantly between groups; specifically, the ADHD-L group performed poorly compared with the TD and ADHD-H groups (*p* < 0.001 for both).

Significant between-group differences were observed in the DBDRS and K-CPT2 scores. Regarding the DBDRS, the participants with ADHD-L were given higher scores than those in the group with TD in the inattentive and hyperactive dimensions by both parents (*p* = 0.001, *p* = 0.037, respectively) and teachers (all *p* < 0.001). The participants with ADHD-H were given higher scores than those in the group with TD on the hyperactive dimension by parents (*p* = 0.005) and on the inattentive and hyperactive dimension by teachers (*p* < 0.001, *p* = 0.001, respectively). Regarding the K-CPT2, the ADHD-L group performed poorly compared with both the TD and ADHD-H groups in detectability (*p* = 0.002, *p* = 0.030, respectively), omission (*p* = 0.007, *p* = 0.016, respectively), HRT (*p* < 0.001 in both), HRT SD (*p* = 0.001, *p* =0.042, respectively), and HRT ISI changes (*p* = 0.001, *p* = 0.028, respectively) (Table 1 & Figure 5).

### 3.2. Resting Relative Spectral Power

Resting relative spectral power did not differ between the groups (Appendix A).

### 3.3. Slow-Task-Related Relative Spectral Power

The PSD on five brain regions in the slow-rate task condition of the two groups (all ADHD versus TD) as well as the three groups (ADHD-H versus ADHD-L versus TD) were compared (Appendix A). Table 2 and Figure 6 present the significant results from the analysis of between-group differences in the slow-rate task-related PSD. The ADHD-H group had significantly higher delta and lower beta values than the TD group in the central region (*p* = 0.008, *p* = 0.018, respectively). The ADHD-L group had significantly higher delta values than the TD group on the Pz electrode (*p* = 0.006).

### 3.4. Fast-Task-Related Relative Spectral Power

The PSD on five brain regions in the fast-rate task condition of the two groups (all ADHD versus TD) as well as the three groups (ADHD-H versus ADHD-L versus TD) were compared (Appendix A). Table 3 and Figure 6 present the significant results from the analysis of between-group differences in the fast-rate task-related PSD. A significance in which the ADHD-L group had a significantly higher value of delta and trending lower values of beta on Pz than the TD group was observed (*p* = 0.011, *p* = 0.039, respectively). Notably, the ADHD-L group had a higher value of delta and theta as well as lower values of alpha and beta on Pz than the ADHD-H group (*p* = 0.011, *p* = 0.031, *p* = 0.008, *p* = 0.002, respectively), which indicated apparent ADHD heterogeneity.

### 3.5. EEG Spectral Power Alterations of Groups under Three Conditions

Notably, the condition and group had a significant interaction effect (*p* = 0.027), and the post hoc test indicated that the ADHD-L group experienced an enhancement in delta power during the shift from the rest to the fast-rate task condition (*p* = 0.001). An enhancement in delta power was observed in the ADHD-H group from rest to the slow-rate task conditions (*p* =0.002). The aforementioned results were distinct from those obtained for the TD group, which manifested no obvious differences in the alteration patterns. Figure 7 presents the alteration patterns of the three groups under the three conditions.

Figure 8 demonstrates the concept of the potential clinical application according to the aforementioned findings of the present study.

## 4. Discussion

To the best of our knowledge, this study is the first to construct a stepwise approach to obtain EEG profiles in preschoolers with ADHD of varied cognitive proficiency by using wireless EEG headsets under various task conditions; specifically, slow-rate and fast-rate task conditions. First, we observed that slow-rate task-related neural dynamics can distinguish ADHD (both ADHD-H and ADHD-L) from TD, which can facilitate clinical diagnosis. Subsequently, fast-rate task-related neural dynamics reflect the heterogeneity in the neurobehavioral functions (e.g., working memory and processing speed) of children with ADHD. The total average delta power alteration patterns of the three groups were distinct. The research results contribute to the understanding of task-related neural dynamics, particularly concerning preschool ADHD. The EEG profiles in the present study may involve specific neural biomarkers that can assist clinical planning for preschoolers with ADHD. As displayed in the Figure 8, in some cases in the absence of complete psychological tests due to individual factors, e.g., lack of time and poor cooperation of children, clinicians may refer to the EEG profiles obtained from this easy-to-use and efficient system to facilitate diagnosis.

Notably, differences in centroparietal delta power under the slow-rate task conditions were observed between the two ADHD groups and the TD group, which is partially inconsistent with the findings of previous studies that have indicated that the resting frontocentral theta power and TBR of children with ADHD are different from those of their TD peers [11,12,14,16]. This discrepancy can be attributed to between-study differences in the ages of the groups and the EEG measurement conditions. Some studies have proposed that an increase in the power of the delta band reflects the maturation lag of the brain in young children with ADHD [51,53,54]. Furthermore, because brain attention dysfunction is usually more evident during cognitive tasks, electrophysiological recordings conducted during task conditions can act as discriminants for the diagnosis of ADHD [55]. Attention functions involve several brain areas, and altered parietal activity in ADHD may indicate abnormalities in the posterior visual attention system that affect performance monitoring, attention reallocation, and visuospatial attention, as well as motor mapping within the peripersonal space [56,57]. We infer that the K-CPT involves visual-input and motor-response tasks that require respondents’ visual attention. This phenomenon is completely different from that observed in the resting condition. Parent behavioral training (PBT) interventions show strong evidence of effectiveness in the treatment of preschoolers at risk for ADHD [2,3,5,58]. On the basis of EEG profiles, PBT should involve slow task training, such as delayed conditioning practice, for children with ADHD of various severities.

In the fast-rate task condition, compared with the ADHD-H group, the ADHD-L group manifested a higher parietal delta and theta power as well as lower alpha and beta power. These results imply ADHD heterogeneity and can be explained by the fact that children with ADHD with low cognitive proficiency are less able to process rapidly presented stimuli than children with ADHD with high cognitive proficiency. On the basis of the hypothesis that ADHD results from neurodevelopmental or neurocognitive deficits, various early interventions have been developed to target these presumed core deficits [3]. Specialists have employed computer-driven, play-based, and exercise programs in early intervention to enhance cognitive ability in young children with ADHD [3,59,60,61,62,63,64]. Therefore, according to these EEG profiles, in addition to behavioral training, mental processing speed (particularly on fast tasks) tailored to the individual should constitute a crucial component of cognitive training for young children with ADHD of low cognitive proficiency [65].

In this study, both ADHD groups manifested an overall enhancement in delta power in the shift from rest to task conditions. These results are in line with those of a study by Rommel et al. [66], which reported the ADHD group showed an increase in delta power from the rest to the task condition, and no significant change in delta power was seen in the control group. Our findings suggest that preschoolers with ADHD with both high and low cognitive proficiency have difficulty maintaining alertness and performing stable executive functions during shifts to task conditions. The EEG alteration profiles indicate that PBT should be performed through approaches such as practice repetition, strategy guidance, and task prompting to enhance the ability of all children with ADHD, regardless of cognitive proficiency, to adjust to task shifting.

In this present study, no significant differences in K-CPT and fast-rate EEG PSD values were observed between the group with ADHD-H and TD. These findings are compatible with Sjöwall’s finding [34], which indicated that a substantial proportion of preschool children with ADHD did not have neuropsychological deficits in any domain. Furthermore, some studies have reported that EEG-based classification failed for ADHD [42,67,68,69], which also supported the evidence of heterogeneity in ADHD.

The wearable wireless semidry-electrode EEG recording system used in the present study has none of the drawbacks of conventional wet-electrode EEG systems. Specialists typically find it challenging to monitor the attention status of young children during task performance due to poor compliance and unnatural situations. In our experiences using the semidry-electrode system, children did not need to endure the lengthy preparation procedure involved in wet-electrode systems (which exacerbates impatient and fidgety behavior in children). Notably, the participants noted that wearing the system was comfortable, similar to wearing stylish headphones. They also mentioned that the wire-less design made task performance feel easy and free; thus, the adopted system promoted the cooperation of the participants in the measurements. Most importantly, the data collected using the adopted system are equivalent in quality to that produced by wet-electrode systems. In summary, the adopted system is highly suitable for clinical use in young children.

The relatively small sample size used is a limitation of this study. Nevertheless, we believe that our findings are clinically relevant to the understanding of the underlying mechanisms of ADHD, particularly preschool ADHD, and can serve as a reference for clinicians and researchers with regard to the potential neural biomarkers.

## 5. Conclusions

The EEG profiles in the present study suggest a stepwise approach; EEG dynamics in the slow-rate task condition and in the shift from rest to the task conditions were disparate in children with ADHD of both high and low cognitive proficiency, which assisted in distinguishing ADHD from TD. EEG dynamics under the fast-rate task conditions were different between children with ADHD with high and low cognitive proficiency, which revealed neurobehavioral heterogeneity in ADHD. The acquisition of EEG profiles by employing the novel wireless system is convenient and efficient for clinical use in children. These profiles can provide clinicians and researchers with information on potential neural markers that aid in the planning of early interventions for preschool ADHD, including individual training and educational programs. Future studies should attempt to use the aforementioned system to measure brain activity and provide real-time neurofeedback during game therapy according to the selected neural biomarkers. This system can also be applied to the various attention-related tasks in which children engage, such as reading and writing.

## Figures and Tables

**Figure 1 jpm-12-00731-f001:**
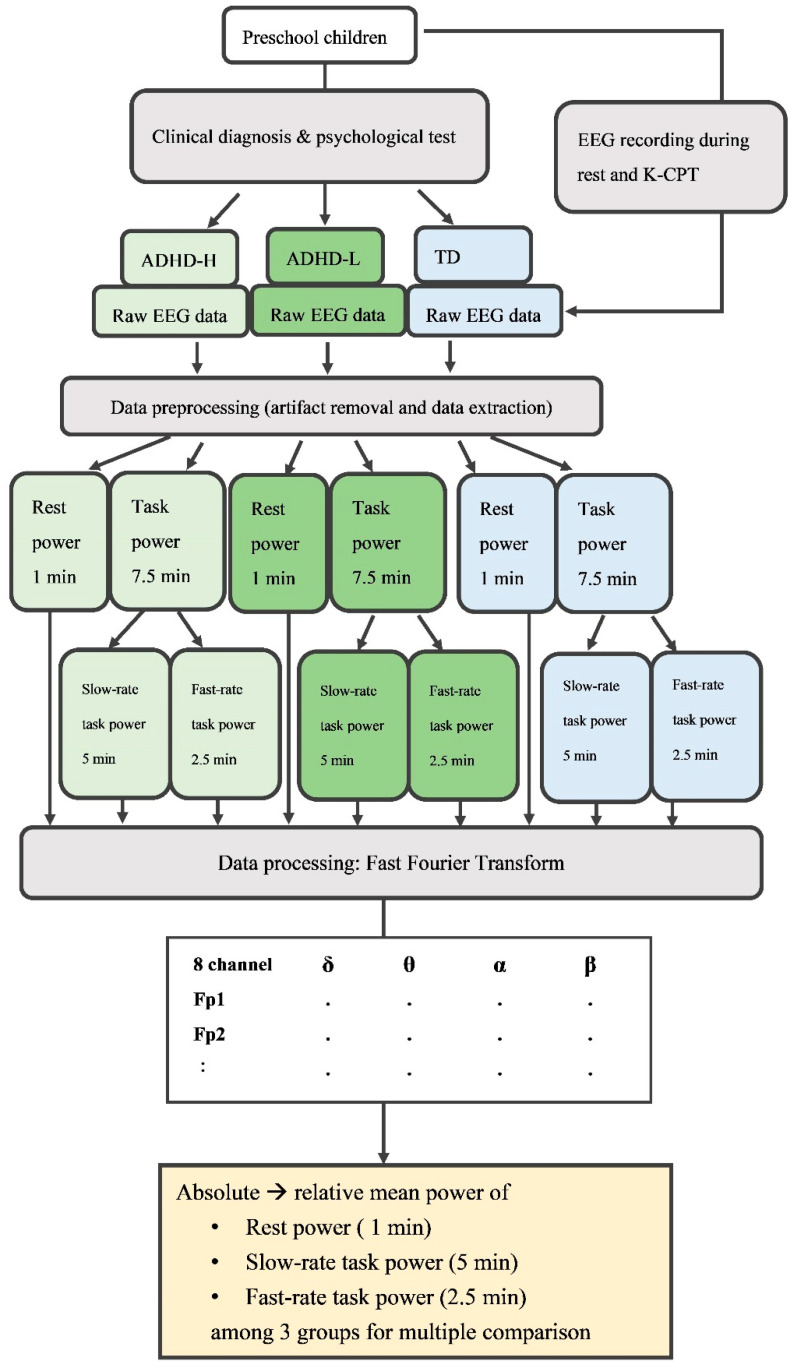
The framework of the study.

**Figure 2 jpm-12-00731-f002:**
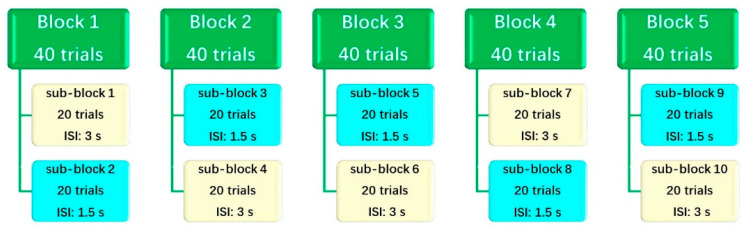
The structure of K-CPT.

**Figure 3 jpm-12-00731-f003:**
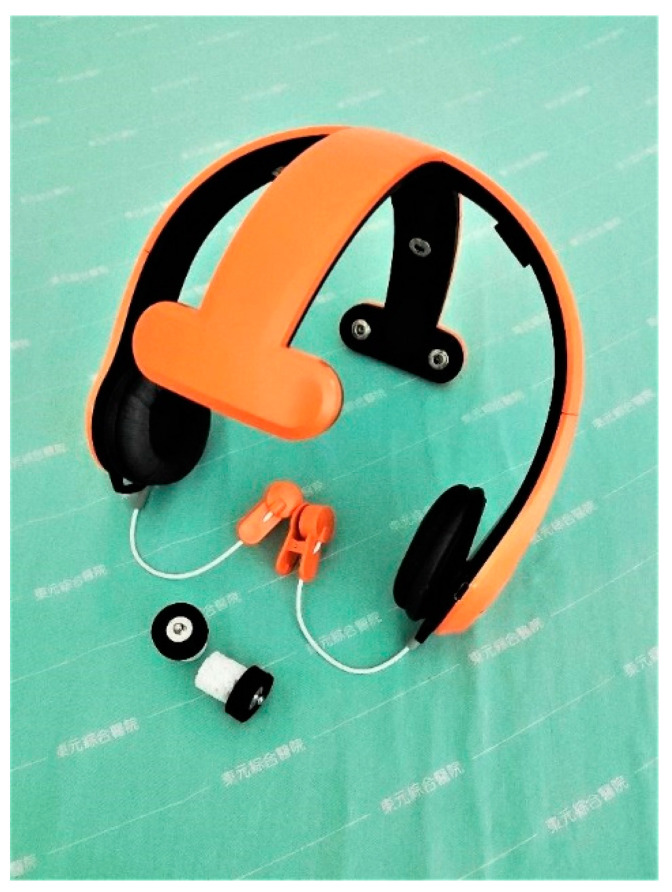
Wireless EEG device. A wireless and wearable EEG headset (**top**) and novel hygroscopic sponge electrodes (**bottom**). The sensors can be easily and quickly assembled, and fit appropriately into the original device.

**Figure 4 jpm-12-00731-f004:**
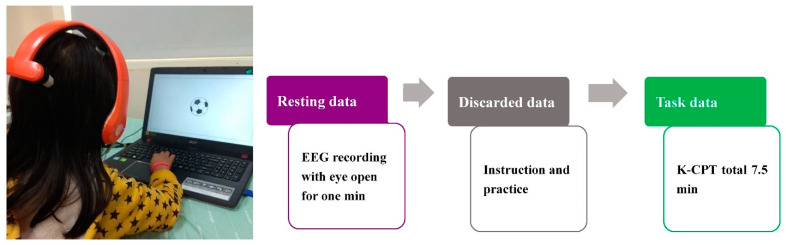
EEG recording procedure.

**Figure 5 jpm-12-00731-f005:**
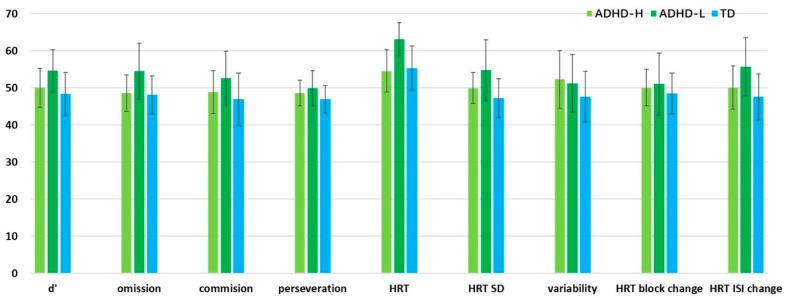
KCPT score among groups.

**Figure 6 jpm-12-00731-f006:**
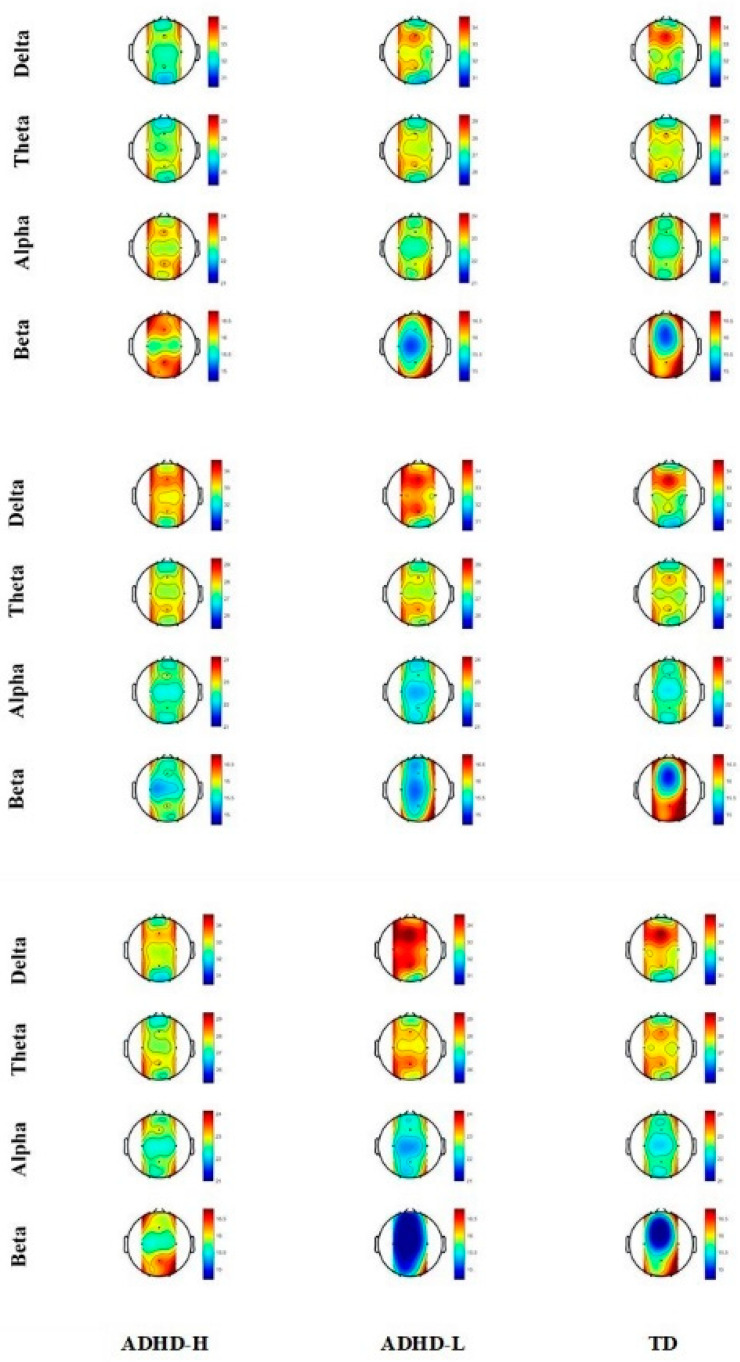
Topography of relative power in three conditions: rest (**upper**), slow-rate task (**middle**), and fast-rate task (**bottom**).

**Figure 7 jpm-12-00731-f007:**
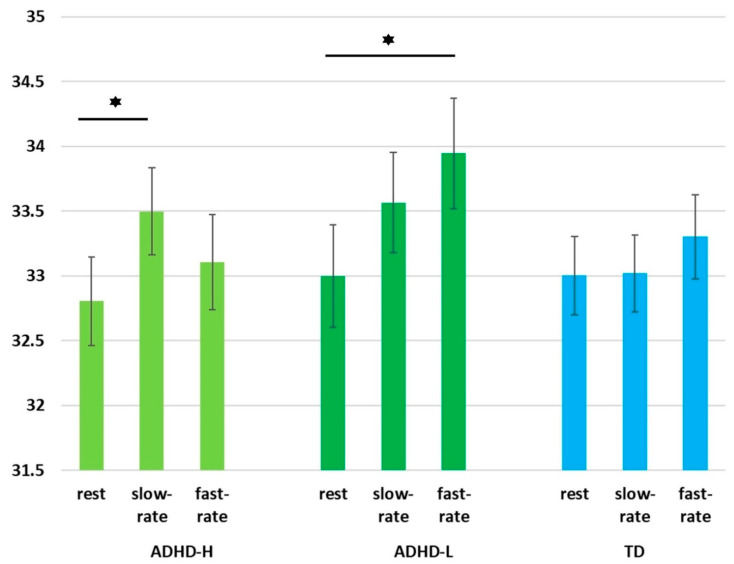
Relative delta power alterations among groups under three conditions (rest, slow-rate, and fast-rate task). * indicated significance, *p* <0.01.

**Figure 8 jpm-12-00731-f008:**
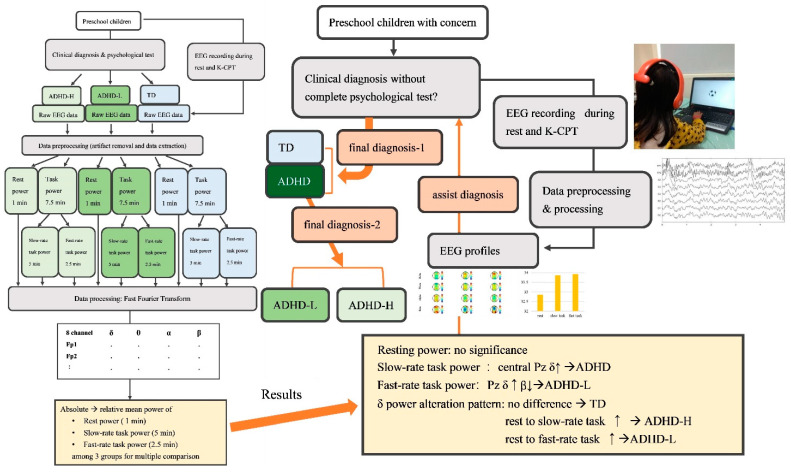
The concept of clinical application.

**Table 1 jpm-12-00731-t001:** Demographic characteristics and K-CPT values of the participants.

Mean ± S.D.	ADHD-H (*n* = 24)	ADHD-L (*n* = 18)	TD (*n* = 31)	*p*	ADHD-H vs. TD (*p*)	ADHD-L vs. TD (*p*)	ADHD-H vs. ADHD-L (*p*)
Age	67.21 ± 5.79	69.00 ± 6.80	67.81 ± 5.09	0.607			
Sex (male: female)	18:6	16:2	23:8	0.482			
FSIQ	99.83 ± 12.24	87.94 ± 13.84	97.66 ± 14.06	**0.015 ***	0.558	**0.019 ***	**0.006 ****
VCI	99.88 ± 12.01	96.11 ± 14.07	98.34 ± 14.21	0.670	0.682	0.582	0.373
CPI	99.76 ± 9.90	75.55 ± 6.96	97.18 ± 13.38	**<0.001 ****	0.465	**<0.001 ****	**<0.001 ****
DBRS-P-i	12.88 ± 3.89	14.89 ± 4.85	10.48 ± 4.68	**0.005 ****	0.054	**0.001 ****	0.154
DBRS-P-h	13.38 ± 5.51	12.50 ± 6.51	8.90 ± 5.34	**0.012 ***	**0.005 ****	**0.037 ***	0.624
DBRS-T-i	13.92 ± 5.37	17.72 ± 4.28	8.47 ± 5.43	**<0.001 ****	**<0.001 ****	**<0.001 ***	**0.021 ***
DBRS-T-h	12.63 ± 5.88	14.00 ± 5.89	6.60 ± 7.24	**<0.001 ****	**0.001 ****	**<0.001 ****	0.499
d’	50.00 ± 6.43	54.61 ± 7.92	48.39 ± 6.08	**0.009 ****	0.378	**0.002 ****	**0.030 ***
omission	48.58 ± 6.11	54.56 ± 9.26	48.13 ± 7.92	**0.016 ***	0.830	**0.007 ****	**0.016 ***
commission	48.88 ± 7.74	52.61 ± 9.68	46.94 ± 7.94	0.078	0.395	**0.025 ***	0.155
perseveration	48.63 ± 5.34	49.89 ± 7.44	46.97 ± 3.83	0.179	0.262	0.072	0.455
HRT	54.58 ± 7.19	63.17 ± 7.31	55.32 ± 6.46	**<0.001 ****	0.695	**<0.001 ****	**<0.001 ****
HRT SD	49.96 ± 6.05	54.78 ± 10.62	47.19 ± 6.13	**0.004 ****	0.177	**0.001 ****	**0.042 ***
variability	52.29 ± 10.15	51.22 ± 9.05	47.65 ± 6.70	0.116	**0.049 ***	0.162	0.689
HRT block change	50.08 ± 6.46	51.11 ± 11.04	48.48 ± 5.80	0.428	0.442	0.247	0.666
HRT ISI change	50.08 ± 6.59	55.67 ± 10.84	47.55 ± 7.03	**0.004 ***	0.248	**0.001 ****	**0.028 ***

Significant differences are indicated in bold ** *p* < 0.01, * *p* < 0.05. K-CPT, Conners Kiddie Continuous Performance Test; SD, standard deviation; ADHD, attention deficit hyperactivity disorder; ADHD-H, ADHD with high cognitive proficiency; ADHD-L, ADHD with low cognitive proficiency; TD, typical development; FSIQ, full-scale intelligence quotient; VCI, verbal comprehension index; CPI, cognitive proficiency index; DBDRS-P-i, Disruptive Behavior Disorders Rating Scale parent version inattentiveness dimension; DBDRS-P-h, Disruptive Behavior Disorders Rating Scale parent version hyperactivity dimension; DBDRS-T-i, Disruptive Behavior Disorders Rating Scale teacher version inattentiveness dimension; DBDRS-T-h, Disruptive Behavior Disorders Rating Scale teacher version hyperactivity dimension; d’, detectability; HRT, hit reaction time; ISI, interstimulus interval.

**Table 2 jpm-12-00731-t002:** Between-group differences in the PSD of significant frequency bands under slow-rate task conditions.

			**Mean**	**SD**	**Overall (*p*)**	**ADHD-H vs. TD (*p*)**	**ADHD-L vs. TD (*p*)**	**ADHD-H vs. ADHD-L (*p*)**
Delta	Ctrl	ADHD-H	33.73	1.38	**0.026**	**0.008 ***	0.145	0.325
		ADHD-L	33.33	1.16				
		TD	32.77	1.28				
	Pz	ADHD-H	33.32	0.81	**0.022**	0.281	**0.006 ***	0.083
		ADHD-L	33.97	1.31				
		TD	32.97	1.35				
Alpha	Pz	ADHD-H	22.65	0.51	0.054	0.895	0.031	0.030
		ADHD-L	22.30	0.45				
		TD	22.63	0.55				
Beta	Ctrl	ADHD-H	15.60	1.72	0.059	**0.018 ***	0.340	0.233
		ADHD-L	16.20	1.64				
		TD	16.66	1.48				
	Pz	ADHD-H	15.96	1.28	0.101	0.286	0.034	0.267
		ADHD-L	15.41	1.57				
		TD	16.41	1.73				

* Significant differences are indicated in bold. *p* < 0.05, the *p*-value was adjusted using a false discovery rate control. PSD, power spectral density, SD, standard deviation; ADHD, attention deficit hyperactivity disorder; ADHD-H, ADHD with high cognitive proficiency; ADHD-L, ADHD with low cognitive proficiency; TD, typical development.

**Table 3 jpm-12-00731-t003:** Between-group differences in the PSD under significant frequency under fast-rate task conditions.

			Mean	SD	Overall (*p*)	ADHD-H vs. TD (*p*)	ADHD-L vs. TD (*p*)	ADHD-H vs. ADHD-L (*p*)
Delta	Pz	ADHD-H	32.92	1.30	**0.002**	0.208	**0.011 ***	**0.001 ***
		ADHD-L	34.27	0.96				
		TD	33.34	1.25				
Theta	Pz	ADHD-H	27.95	1.09	0.090	0.154	0.327	**0.031 ***
		ADHD-L	28.57	0.80				
		TD	28.30	0.79				
Alpha	Pz	ADHD-H	22.76	0.79	**0.030**	0.195	0.102	**0.008 ***
		ADHD-L	22.26	0.39				
		TD	22.55	0.50				
Beta	Pz	ADHD-H	16.37	1.81	**0.007**	0.161	0.039	**0.002 ***
		ADHD-L	14.90	0.66				
		TD	15.81	1.48				

* Significant differences are indicated in bold. *p* < 0.05, the *p*-value was adjusted using a false discovery rate control. PSD, power spectral density, SD, standard deviation; ADHD, attention deficit hyperactivity disorder; ADHD-H, ADHD with high cognitive proficiency; ADHD-L, ADHD with low cognitive proficiency; TD, typical development.

## Data Availability

The data presented in this study are available upon request from the corresponding author.

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
