# Peer review of "Task-Rate-Related Neural Dynamics Using Wireless EEG to Assist Diagnosis and Intervention Planning for Preschoolers with ADHD Exhibiting Heterogeneous Cognitive Proficiency"

_jpm, 2022, doi:10.3390/jpm12050731_

Round 1
Reviewer 1 Report
The manuscript entitled “Task-rate-related Neural Dynamics via using Wireless EEG to Assist Diagnosis and Intervention Planning for Preschoolers with ADHD of Heterogeneous Cognitive Proficiency” by Chen et al. has been carried out to find a specific EEG signal or pattern in different samples of ADHD and typically developing children. The results of the present study have significant value for clinical of ADHD, specifically if we are trying to find biomarkers. The manuscript is well organized and written. I have included below some considerations.
- Authors used K-CPT2 in order to measure sustained attention; however, the test used is aimed to measure perception process as well. Do the authors consider perception was similar in the sample? This issue should be cleared.
- Authors included a tabled to show the results in K-CPT2. I consider a graphic could help to readers; that is, a graphic is more direct, and it usually makes easy to catch the different performance between groups.
- If I understood the manuscript correctly, authors used in the cognitive task a procedure similar to signal detection theory. Could authors include the strategy used by group? They have detectability index, so it could be interesting to analyse how participants performed the task.
- Minor points: Figure 2: it is difficult to read it. Could authors make it bigger?
Reviewer 2 Report
The authors have produced an interesting paper on the evaluation of preschool children with ADHD by EEG and have provided a method that could offer new perspectives of evaluation for these children, which makes their work very interesting. However, there are some issues that should be clarified in more detail.
1.- Paragraph 56-67 mentions one of the most studied findings in the literature regarding ADHD: the elevated Theta/Beta ratio. but there are other common findings that could be mentioned. For example, these mentioned by Clarke (2002): https://doi.org/10.1016/S1388-2457(01)00668-X
2.- Paragraph 185-198 states that the impedances obtained were less than 100KOhm. Impedances of less than 5KOhm are usually recommended in the EEG literature. The authors seem to give an explanation for this, but could they elaborate on it, did the authors control the KOhm differences between the different sensors, and if so, what were they?
3.- The authors established fixed frequency bands, as usual, but did they check that their participants had an age-expected Alpha Peak Frequency (APF)? Some authors have suggested that the could be a biomarker of response to different treatments and that it could be related to the different bandwidths of EEG components. https://www.tandfonline.com/doi/abs/10.1080/10874208.2012.677664
https://www.mdpi.com/2076-3425/11/2/167
https://www.tandfonline.com/doi/abs/10.1300/J184v05n01_08
The work presented by the authors is very interesting both from a scientific and clinical point of view. Congratulations
